# Recent Progress of 3D Printing of Polymer Electrolyte Membrane-Based Fuel Cells for Clean Energy Generation

**DOI:** 10.3390/polym15234553

**Published:** 2023-11-28

**Authors:** Sergey S. Golubkov, Sofia M. Morozova

**Affiliations:** 1Moscow Institute of Physics and Technology, National Research University, Institutskiy per. 9, 141700 Dolgoprudny, Russia; golserg97@yandex.ru; 2N.E. Bauman Moscow State Technical University, 2nd Baumanskaya St. 5/1, 105005 Moscow, Russia

**Keywords:** 3D printing, additive manufacturing, fuel cell, energy conversion, sustainable energy, polymer electrolyte membrane

## Abstract

This review summarizes recent advances in the application of 3D printing (additive manufacturing) for the fabrication of various components of hydrogen fuel cells with a polymer electrolyte membrane (HFC-PEMs). This type of fuel cell is an example of green renewable energy, but its active implementation into the real industry is fraught with a number of problems, including rapid degradation and low efficiency. The application of 3D printing is promising for improvement in HFC-PEM performance due to the possibility of creating complex geometric shapes, the exact location of components on the substrate, as well as the low-cost and simplicity of the process. This review examines the use of various 3D printing techniques, such as inkjet printing, fused deposition modeling (FDM) and stereolithography, for the production/modification of electrodes, gas diffusion and catalyst layers, as well as bipolar plates. In conclusion, the challenges and possible solutions of the identified drawbacks for further development in this field of research are discussed. It is expected that this review article will benefit both representatives of applied science interested in specific engineering solutions and fundamental science aimed at studying the processes occurring in the fuel cell.

## 1. Introduction

In recent years, humanity has been paying more and more attention to alternative energy sources, such as solar panels, wind generators and electrochemical energy storage [1,2]. Fuel cells are one of the most promising solutions, because they do not depend on weather conditions, unlike solar panels and wind turbines; they have greater efficiency than internal combustion engines and are more environmentally friendly than lithium batteries [3,4]. Compared with other types of hydrogen fuel cells (HFCs) such as a solid-state oxide one based on molten carbonate, orthophosphoric acid and alkaline, polymer electrolyte membrane (PEM) fuel cells are of particular interest due to their ability to operate at low temperatures (<100 °C) [5,6]. The structure of the main part of the HFC-PEM is a membrane-electrode block (MEB), where the conversion of hydrogen and air oxygen into electricity occurs. The detailed structure of an MEB is schematically presented on Figure 1a and includes a polymer membrane, porous electrodes with a catalyst layer, a gas diffusion layer and bipolar plates.

The reaction of hydrogen at the cathode occurs with the release of two electrons and two hydrogen ions, which, passing through an electrical circuit and a proton-conducting membrane, correspondingly, react with oxygen on an anode to form water (Figure 1a). Improvement in the individual elements of a fuel cell, for example, a membrane, a catalytic layer, a gas diffusion layer, etc., allows for improving the parameters of the fuel cell, such as the energy density, maximum current, degradation time and ohmic losses [7].

Due to the peculiarities of fuel cells, many elements for their manufacture must have a complex geometric shape or a micro- and nanostructure [8]. For example, bipolar plates provide a constant certain current of hydrogen and the removal of reaction products and heat due to a complex system of channels, the optimization of which is still being discussed [9,10]. The gas diffusion layer located between the bipolar plates and the catalyst deposited to the electrode (Figure 1a) should have a complex microstructure, i.e., porosity for providing sufficient permeation of gases and help for water vapor to reach the membrane and enhance its ionic conductivity [11,12]. The deposition of catalyst layers in fuel cells requires thickness and uniformity control for high fuel cell performance [13,14]. Traditional inexpensive methods of manufacturing fuel cell parts with a complex structure are milling, molding and dry pressing [15]. The advantage of 3D printing in comparison with these methods is a greater resolution when creating a complex structure and the easiness of changing the structure by creating a digital code, rather than making new molds/templates, such as when pressing (Figure 1b). Chemical and physical vapor deposition, spray coating and electrospinning are used to create thin nanoscale or nanostructured coatings. Compared to these methods, 3D printing is cheaper and more scalable (Figure 1b). Moreover, 3D printing allows for significantly increasing the speed of component production compared to conventional techniques and also allows for making mobile (portable) production centers that are not tied to a specific location, i.e., a factory. The advantages of 3D printing are of a general nature, which is confirmed by its active introduction to produce not only fuel cell components but also conductive hydrogels and membranes for wearable electronics and sensors [16,17].

Thus, 3D printing techniques are mainly relevant as an alternative to the methods of manufacturing fuel cell components, which is reflected in the recently published reviews [18,19,20]. While previously published reviews describe the 3D printing of membranes [18] for different applications or 3D printing not only fuel cells with a polymer electrolyte membrane but including ceramic membranes and liquid electrolytes [17,19,20], in this review we focus on the recent progress made in the application of 3D printing for the fabrication of fuel cells with a polymer electrolyte membrane. The purpose of this review is to focus on PEM-HFCs as the most promising for the mass production of civil transport (cars and boats) or surveillance drones. Mass application requires cheaper and simpler production technology, which makes 3D printing techniques extremely promising. Section 1 discusses the features of the different 3D printing techniques, Section 2 is devoted to printing certain elements and Section 3 reveals the improvements that were caused by the utilization of 3D printing. In conclusion, future directions of development of the field and ways of overcoming challenges are discussed.

## 2. Three-Dimensional Printing for Fuel Cell Fabrication

### 2.1. Overview of 3D Printing for Fuel Cell Fabrication

Additive technologies, including 3D printing, have been actively developed in recent years and are utilized in construction [21], the food industry [22], medicine [23], optics [24], electronics [25] and energetics [26]. There are many types of 3D printing techniques; however, in this review, we focus only on those that are utilized for the manufacture of HFC-PEM elements, namely, inkjet printing, fused deposition modeling (FDM), selective laser sintering (SLS) and 3D stereolithography.

*Inkjet printing* consists of spraying liquid inks from nozzles combined in print heads (Figure 2a), and the droplets formed are very small in volume of the order of picoliters [27].

In a “drop-on-demand” printer, ink droplet formation occurs due to the piezoelectric compression of the print head when an electric signal is applied. In order for the ink to be printable, its rheological properties must correspond to the Ohnesorge number to be in the range from 0.1 to 1 [28]. The advantages of this method include a high resolution (up to 10 μm) in the case of a “drop-on-demand”-type printer [29]. However, the method is more time-consuming in producing material compared to other 3D techniques and is more suitable for creating coatings than 3D structures. Another disadvantage is that optimization of rheological parameters often involves the addition of surfactants to control surface tension, which can negatively affect the electrochemical stability of the fuel cell [30]. Due to these features, inkjet printing has found application mostly for depositing catalytic layers to form electrodes for HFC-PEMs (Figure 2a′).

In the *FDM* technique, there is an alternate deposition of layers of slurry materials, such as melted thermoplastics, viscous polymer solutions or shear-thinning gels [31]: the fixation of the printed form may be associated with the formation of a melt or the introduction of crosslinking additives, including photopolymerization (Figure 2b) [32]. The requirements for the rheological characteristics for FDM printing is the viscosity of inks in the range of 10^2^–10^9^ Pa·s and the ability to quickly fix the shape after printing using a chemical reaction (photopolymerization, a reaction with a crosslinking agent) or a physical method (a phase transition and thixotropic behavior) (Figure 2b). The advantages of the method include efficiency, good equipment performance, a large range of raw materials that can be used, the ability to produce complex parts and elements with a flexible or soft structure, as well as shock- and vibration-resistant products [33]. The disadvantage is the layered structure of the finished objects, which could not possess enough tough mechanical properties for some fuel cell parts. Also, this technique does not have such a high print resolution (up to 100 μm) as inkjet printing or stereolithography (Figure 2). FDM was used for the production of flexible bipolar plates [34] and GDLs [35], catalyst deposition [36] and the patterning of an electrolyte membrane [37] (Figure 2b′).

During the *SLS* process, the laser beam melts the metal or ceramic [38], and rarely polymer [39] powder, after which the layer is re-applied and processed (Figure 2c). The highest resolution is up to 50 μm, but that extremely depends on the viscosity of the melted precursors. An advantage of selective melting is the possibility of manufacturing parts with a complex configuration, such as non-standard dimensions that cannot be created by injection molding or milling [38]. The disadvantages are the high cost of materials and equipment, the need for sandblasting the surfaces of the desired parts and the applicability only for metal or ceramic-based elements. Although the printing resolution is higher than that of moldings or milling, it is still lower compared to other 3D printing techniques, such as 3D stereolithography and inkjet printing [38]. Thus, SLS is used for the fabrication of rather massive elements such as GDL [40] or bipolar plates [41] for HFC-PEMs (Figure 2c′).

*Stereolithography* uses a liquid precursor, most commonly a photopolymer, which could be photopolymerized [42]. A laser, projector or LCD monitor is used for the directional supply of raw materials that harden upon irradiation. The printing platform is raised, which allows for repeating the operation to create the desired product in layers. The main advantage of the technology is the high resolution of the manufactured product (the minimum layer thickness is 20 µm, and the XY resolution level starts from 50 µm, which is typical for popular models of three-dimensional printers). Depending on the characteristics of the material used, such printing allows for producing both solid and flexible elements; good smoothness of the surfaces ensures minimal processing efforts after production. The rheological characteristics of stereolithography ink include limitations for viscosity, i.e., the viscosity should be no more than 10^3^ Pa·s, so that the material can flow freely from the hardened layers without disturbing the print resolution. The disadvantages of the technology include a relatively low performance indicator and a fairly high price for the equipment used. Due to the specific requirements for the precursors (the ability to be photopolymerized), this method is not frequently utilized; however, some examples related to the patterning of a PEM or bipolar plates production are known [20].

### 2.2. Components of Fuel Cells Fabricated by 3D Printing

Three directions play an important role in the production of HFC-PEM components: (i) the formation of a micro- or nanostructure, (ii) the production of components with thinner and miniaturized parts to reduce the weight of fuel cells as a whole and achieve good compactness and (iii) increasing the speed and reducing production costs. All this has made the use of 3D printing an important strategy in the design and assembly of HFCs. This section describes the progress made and the features of the application of 3D printing for each of the components.

#### 2.2.1. Bipolar Plates

Bipolar plates, or so-called flow field plates, account for up to 80% of the total weight and approximately half of the inventory cost value in HFC-PEMs [43]. Bipolar plates are used to provide a stable and good interconnection between other cell components, minimize electrical resistance, transfer reagents and by-products and maintain heat efficiency [44]. There are two points in bipolar plates: (i) material requirements and (ii) channel design. Therefore, 3D printing is in demand for the manufacture of bipolar plates because it allows for the utilization of inexpensive and lightweight materials and performs complex geometry.

The first attempts were connected to printing graphite [45] or metal alloy [46] as chemically resistant materials. However, this approach has a disadvantage associated with the fragility of the resulting bipolar plates. Recent works are aimed at the use of polymers or polymer composites [47], which not only solve the problem of fragility but also introduce new properties, for example, flexibility [34]. For example, flexible bipolar plates were fabricated by an FDM-like technique based on jetting with the following UV fixation (Figure 3a) of a flexible and translucent photopolymer called TangoPlus (with a tensile strength of 0.8–1.5 MPa and a breaking elongation of 170–220%) [34]. It was shown that an assembled fuel cell could work both in the flat and bent positions (Figure 3a′) and demonstrates a power density of 30.2 and 87.1 mW·cm^−2^, respectively (Figure 3a″).

For the channel design, the trend is to use bio-inspired structures, such as leaf veins, lungs, trees or unconventional shapes, which, due to their complex geometry, are easier to obtain using additive technologies [48]. In comparison with the conventional flow channels, the bio-inspired one has a more efficient internal mass process and heat transfer, which improves the overall performance of HFC-PEMs. Mostly, these structures have a fractal nature, i.e., self-similarity, and the “generation number” or “generation” is used to describe them, i.e., the number of repetitions of self-similarity [49]. In recent work that was 3D printed by a metal laser-sintering lung-inspired bipolar plate with four generations showed up to a 30% increase in performance (power density) with respect to a conventional serpentine flow field (at current densities higher than 0.8 A cm^−2^ and 75% relative humidity (RH)) and the lowest voltage decay (∼5 mV·h^−1^) [50]. Despite the fact that the simulation results show that the design with N = 5 provides an optimal PEFC performance, in practice, the use of a fractal flow field above N = 4 is limited by the resolution of 3D printing.

**Figure 3 polymers-15-04553-f003:**
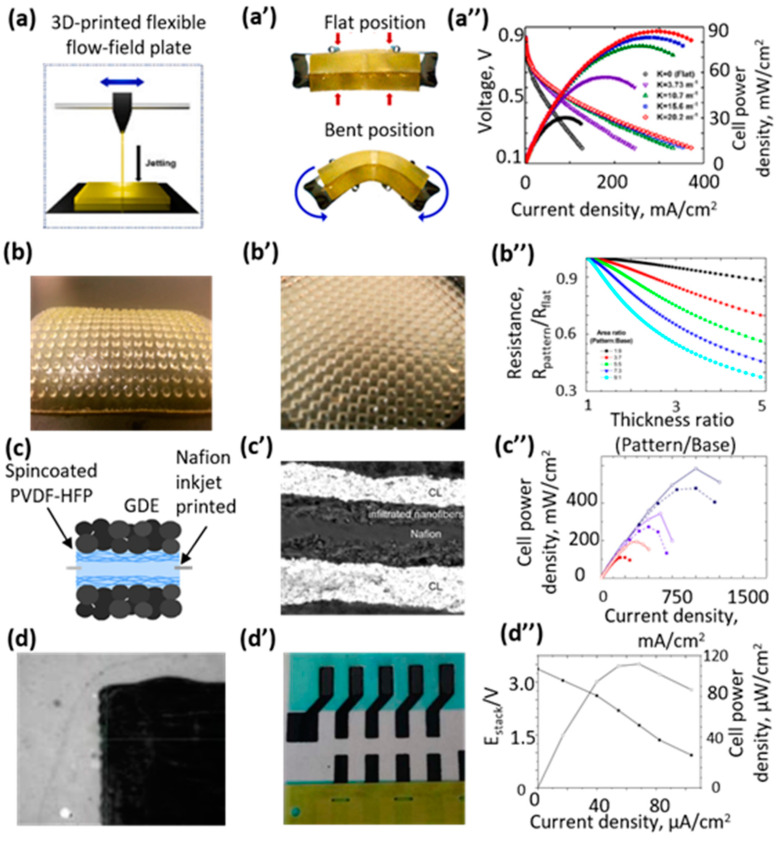
Examples of 3D printing techniques for fabrication of fuel cell elements. (**a**) Scheme of working principle of FDM-like technique based on jetting with following UV fixation. (**a′**) Appearance of 3D printed bipolar plates in flat (top) and bent (bottom) positions. (**a″**) Polarization curves with different curvature (closed symbols represent power densities and open symbols represent voltages). Reproduced with permission from ref. [34], Copyright 2022, Elsevier. (**b**,**b′**) Appearance of membrane patterned by 3D photolithographic process similar to stereolithography (membrane materials are poly(4-vinylbenzyl chloride) crosslinked by dimethacrylate and poly(ethylene glycol) diacrylate). (**b″**) Calculated resistance of patterned membranes compared to flat membranes as functions of thickness and area ratios. Reproduced with permission from ref. [37], Copyright 2016, American Chemical Society. (**c**) Scheme of fabrication of inkjet-printed Nafion-based membrane. (**c′**) SEM image of cryo-fractured cross section of the HFS-PEM, demonstrating complexity of structure of the printed membrane. (**c″**) Polarization data comparing the Nafion-printed membrane fuel cell (solid dots) to the directly deposited composite membrane fuel cell (empty dots). Measurement conditions were 80 °C and 95% RH (black color); 100 °C and 70% RH (blue color); and 120 °C and 35% RH (red color). Reproduced with permission from ref. [51], Copyright 2017, Elsevier. (**d**,**d′**) Appearance of inkjet printing-deposited anode and cathode catalyst layers taken by camera integrated with the printer (**d**) and overview of printed electrodes (**d′**). (**d″**) A polarization curve of HFC with a H_2_ flow of 50 mL min^−1^ in the electrodes at 21 °C; reprinted with permission from ref. [52], Copyright 2020, IOP Publishing.

#### 2.2.2. Gas Diffusion Layer

A GDL functions as a current channel and is located between the bipolar plates and the electrodes and must have sufficient porosity to ensure effective diffusion for reagents, water and heat, as well as protection of the electrode catalyst layers from degradation [53]. The production of a GDL by 3D printing makes it possible to improve the stability of the catalyst and reduce the ohmic losses and brittleness of the layer compared to a GDL produced by traditional methods, such as casting or spraying [33]. Due to the demands on the mechanical properties and chemical stability for laser-based printing techniques, SLS or SLM are more often used for a GDL. For example, a GDL based on Ti_6_Al_4_V with a complex geometric structure was printed for proton exchange membrane electrolyzers. Titanium alloy has been used to solve the problem of electrochemical corrosion/degradation due to its greater resistance compared to carbon materials. The high porosity achieved by 3D printing and the increased corrosion resistance made it possible to achieve low ohmic losses up to 0.21 ohms/cm^2^ at 65 °C [54].

#### 2.2.3. Polyelectrolyte

Polyelectrolyte membrane printing is the least common at the moment [15], although it has a number of advantages, for example, improved contact with the electrodes and uniformity of the catalyst distribution. Also, in a recent review, it was shown that micro- and nanopatterning by different fabrication methods of the membrane’s surface makes it possible to improve the electrochemical parameters of fuel cells [55]. Light-assisted 3D printing is beneficial for the patterning of membranes due to the high resolution of the method. For example, micropatterned anion exchange membranes have been 3D printed via a photoinitiated free radical polymerization of 4-vinylbenzyl chloride crosslinked by dimethacrylate and poly(ethylene glycol) diacrylate (PEGDA) [37]. The appearance of different patterns are shown in Figure 3b,b′. Depending on the pattern and concentration of the PEGDA crosslinker, printed membranes demonstrate a difference in the water uptake, which influences the charge transport. Figure 3b″ shows that patterned membranes have lower ionic resistances in comparison with flat (smooth) membranes at the same parameters (thickness and chemical formulation), which was explained with a parallel resistance model by comparing the resistance of a patterned membrane with that of a flat membrane with an equivalent effective thickness (volume/area) [37].

A proton-conducting membrane was prepared by inkjet printing the Nafion ionomer dispersion into the pore space of a fiber mat based on poly(vinylidene fluoride-co-hexafluoropropylene) (PVDF-HFP) electrospun onto gas diffusion electrodes [51]. A lower ionic resistance and, especially at 120 °C, a reduced charge transfer resistance is found compared to the Nafion HP membrane.

#### 2.2.4. Catalyst Layer

The properties of the electrodes play a key role in the electrochemical process in HFC-PEMs. According to numerous data, the simplest, most effective and frequently used catalyst is Pt on carbon [56]. Due to the high cost of precious metals, which include platinum, the catalytic layer is up to 20% of the cost of the entire fuel cell [57]. Therefore, the reduction in material costs that can be achieved with 3D printing is crucial to stimulate the commercialization of PEMFCs, especially for transport applications. To achieve thin layers and minimize material consumption, inkjet printing is often used. For example, a catalyst based on 2.5 wt% of carbon–platinum–ruthenium was mixed with a 0.5% Nafion concentration in a diacetone alcohol and used as inks for the inkjet printing of an anode for a fully printed flexible fuel cell stack (Figure 3d,d′) [52]. These anodes showed resistivity of 0.1 Ω cm, which is close to the commercial anode resistivity of 0.05 Ω cm, and a fuel cell based on them had an open-circuit potential under H_2_/air conditions of 3.4 V.

## 3. Effect of 3D Printed Components on Fuel Cell Parameters

The main parameters that characterize the fuel cell are the specific capacity, power density, A and open-circuit voltage (OCV) [5]. There is also a group of parameters related to the degradation of the catalyst and membrane that determine the operating time of the element [15]. The most important parameter is the power density, which is the amount of power (the time rate of energy transfer) per unit volume (W/m^3^) and OCV (V). In Table 1, the recent progress in 3D printed HFC-PEMs is summarized.

A catalyst layer was fabricated by 3D microextrusion printing with a mixture of graphite/ethanol/water [36]. The microextrusion method in this study was successfully used to prevent the detachment of the catalyst layer. A fully assembled cell demonstrates a power density of 727 mW/cm^2^ at 80 °C, which was 13% lower in comparison with a non-3D printed standard electrode (829 mW/cm^2^). However, the OCV of the printed element was slightly lower than for a standard electrode (0.980 instead of 0.999 V). The power characteristics for a 3D printed element have not decreased critically, but the utilization of 3D printing allows for increasing the speed of its creation [36].

A commercial 3D printing system and conventional acrylonitrile-butadiene-styrene (ABS) filament were used to manufacture 3D printed polymer bipolar plates [58]. After the preparation of the polymer bipolar plates, Ag current-collecting layers were deposited on top of the bipolar plates using a commercial sputtering system. The difference in the HFC was in the thickness of the current collection layer on the bipolar plates. In summary, when the thickness of the Ag-based current-collecting layer was increased from 216 nm to 1046 nm (384.3%), the maximum power density of the fuel cell was increased from 114.34 mW/cm^2^ to 308.35 mW/cm^2^ (169.7%) because of the reduction in the total ohmic resistance from 0.953 Ω·cm^2^ to 0.258 Ω·cm^2^ (72.9%).

In a recently published paper, an integrated flow-field GDL (i-FF-GDL) was printed by the microextrusion of a TiH_2_ suspension and high-temperature treatment after printing to make a porous and high-conductivity Ti layer. This porous structure was used to improve the gas diffusion and facilitate water drainage. The peak power density of the element increased by 15% and 8% by using an i-FF-GDL under H_2_–O_2_ and H_2_-air (CO_2_-free) conditions compared with the GDL made of commercial carbon paper [35].

Three approaches have been developed for multi-layer production with varying degrees of the use of inkjet printing: (a) the catalytic layer applied to the matrix with slots served as the base layer for the first strategy, while the ionomer layer and the second catalyst layer were sequentially applied by inkjet printing over the surface with slots; (b) another approach was to apply a catalytic layer to a commercial membrane as a substrate; and (c), according to the third method, inkjet printing of all three layers was performed, and a layer of membrane reinforcement (i.e., a PTFE film) was used as the initial substrate [59]. The fuel cell with a printed catalyst layer competed closely and slightly surpassed the fuel element of traditional production in terms of voltage losses and power density for current densities higher than 800 mA/cm^2^: the peak power density was about 15% higher, which might be accounted to a different average porosity. It is worth remarking that at a current density lower than 800 mA cm^2^, both printed and non-printed devices exhibit nearly the same performance, with a negligible difference in the polarization curve. The comparison highlighted the consistency between the two assemblies, with the performance of the inkjet-printed HFC not being penalized by higher resistance or higher mass transport losses locally occurring as a result of defects.

A novel approach to simplify the fabrication of thin composite membranes for proton exchange fuel cells operating in the medium temperature range between 80 °C and 120 °C was presented in this work [51]. Direct electrospinning of PVDF-HFP nanofibers onto gas diffusion electrodes and the subsequent inkjet printing of Nafion into the nanofiber mesh enabled a fast, simple and scalable fabrication of 12 µm thin composite membranes. Both deposition processes, electrospinning and inkjet printing, are scalable to high throughput and therefore suitable for the industrial production of composite membranes. To improve the characteristics of a fuel cell, hot pressing the materials after 3D printing was also used. At 120 °C and 35% relative humidity, with a stoichiometric 1.5/2.5 H_2_/air flow and atmospheric pressure, the power density of the DMD fuel cell (0.19 W cm^−2^) was about 1.7 times higher than that of the reference fuel cell (0.11 W cm^−2^) with a Nafion HP membrane and identical catalyst. A lower ionic resistance and, especially at 120 °C, a reduced charge transfer resistance were found compared to the Nafion HP membrane. A 100 h accelerated stress test revealed a voltage decay of below 0.8 mV h^−1^, which is in the range of the literature values for significantly thicker reinforced membranes [51].

The method of deposition of a catalyst layer by 3D printing directly onto a membrane or cathode GDL was reported [61]. The anode fabrication process was constant for all the experiments and entailed the deposition of 10 layers of ink in square patterns on a carbon paper substrate. The difference between the samples is in the quantity of printed layers on different parts of the cell. The catalyst thickness varied from 1 to 3 micron, depending on the quantity of 3D printed layers. It was shown that printing the catalyst layer on the GDL makes the max power density bigger than the same layers of the catalyst printed on a membrane. In total, 16 layers on a GDL have a performance power density 2.5 times bigger than 16 layers on a membrane. Also, printing on a GDL shows a lower OCV. On average, the use of 3D printing allows for either an increase in the energy density by 20% or an increase in the production speed of components with low losses for power characteristics (not more than 15%).

## 4. Conclusions

It was revealed that design and component fabrication require approximately half of the total cost for a single HFC-PEM [62]. Thus, 3D printing is highly beneficial for reducing the cost and weight of the designed HFC-PEM components; however, several limitations are hampering the commercialization of most of the 3D printing techniques for scalable fuel cell fabrication. For example, the slurry-based techniques, such as FDM, or jetting techniques, such as inkjet printing, require specific viscosity parameters for printing: either low viscosity for photocurable or thixotropic highly viscous materials. Thus, a limited number of suitable materials is an issue too. The additional postprinting processes also require the removal of organic solvents and other components that could be creating a large number of pores and hence weakening the mechanical properties. Although this seriously limits the strength of the resulting materials, it is also an advantage for obtaining highly porous elements, such as a catalytic layer. As a rule, 3D printing technologies based on powder melting/sintering are used to manufacture components with high mechanical properties. However, limitations on the material are also present—the ability to melt, the price and the chemical resistance to oxygen as an oxidizer and hydrogen as a reducing agent. Thus, although 3D printing is actively used for the production of fuel cell components, the examples in most cases reflect the production of bipolar plates and, to a lesser extent, other components (gas diffusion layers, membranes and catalytic layers).

These challenges open up new ways for future field development: (i) a focus on printing membranes as a way to achieve an optimal thickness, uniform distribution of the membrane (good contacts with other components) and a composite nature; (ii) a combination of several printing techniques for different elements, while at the moment only one printed component is being studied in the literature at once; (iii) the active introduction of machine learning and artificial technologies into the printing process to increase the variability of products, optimize printing parameters and for the generation of new effective patterns for bipolar plates and GDLs; and (iv) the creation of mobile (portable) 3D printing stations capable of creating the necessary components of fuel cells and assembling them directly at the production site without need for delivery. Although 3D printing has already established itself as an effective tool for creating fuel cell elements, the potential of the method has yet to be fully revealed.

## Figures and Tables

**Figure 1 polymers-15-04553-f001:**
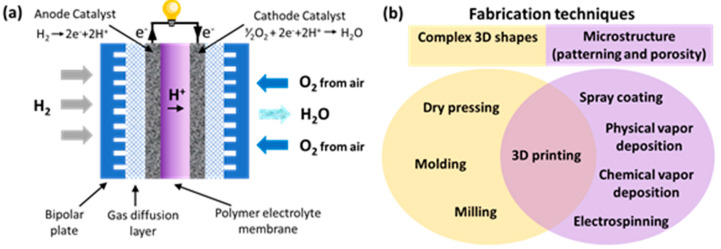
(**a**) Schematic diagram of the operation of a hydrogen-air fuel cell; (**b**) features of different techniques suitable for fabrication of HFC−PEM elements.

**Figure 2 polymers-15-04553-f002:**
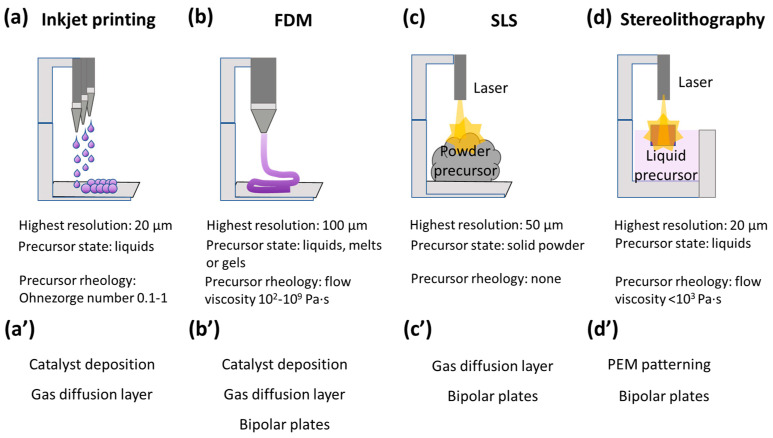
Scheme of the working principles for 3D printing techniques for inkjet printing (**a**), FDM (**b**), SLS (**c**) and stereolithography (**d**). Types of HFC-PEM elements that are beneficial to produce by using 3D printing techniques, such as inkjet printing (**a′**), FDM (**b′**), SLS (**c′**) and stereolithography (**d′**).

**Table 1 polymers-15-04553-t001:** Power density for 3D printed HFC-PEM.

Type of Printing	Printed Element	Other Elements of Fuel Cell	A, mW/cm^2^	OCV, V	T, ^o^C	Power Density of Non-3D Printed Analogue	Ref.
FDM	Catalyst layers	40 wt% Pt/C5, hydroalcoholic Nafion	727	0.98	80	829	[36]
FDM	Bipolar plates	40 wt% Pt/C (40 wt% Pt), Nafion solution 5 wt% in water-alcohol	87.1	ND *	25	30.2	[34]
FDM	Bipolar plates	Nafion 211 commercial membrane	308.35	1.02	25	ND	[58]
FDM	GDL	40 wt% Pt/C, Nafion solution 2 wt% was mixed in ratio 0.25	1200	ND	80	ND	[35]
SLS	GDL	0.5 mg Pt/cm^2^ on either side of the Nafion membrane	0.5	ND	75	ND	[40]
Inkjet printing	Catalyst and membrane layers	Commercial Nafion^®^ 115 membrane (125 mm thickness)	800	0.5	60	~650	[59]
Inkjet printing	Membrane	Nafion D2020 dispersion	190	ND	120	110	[51]
Inkjet printing	Catalyst layers	Nafion ionomer (5 wt%), 50 wt% Pt/C	579	ND	25	ND	[60]
Inkjet printing	Catalyst layers	Nafion ionomer dispersion (4.24 wt% of total, or 0.21 wt% Nafion), 50 wt% Pt/C	550	ND	70	220	[61]

* ND—no data.

## Data Availability

Not applicable.

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
