# Peer review of "Recent Progress of 3D Printing of Polymer Electrolyte Membrane-Based Fuel Cells for Clean Energy Generation"

_polymers, 2023, doi:10.3390/polym15234553_

Round 1
Reviewer 1 Report
Comments and Suggestions for Authors
This text is a review article devoted to 3D printing of polymer electrolyte membrane FC for clean energy generation. It is based on 57 recent articles related to the subject, of which 27 are already reviews, most published between 2021 and 2023.
What is the difference, for the authors, between “Proton Exchange Membrane Fuel Cells” (commonly referred to as PEMFCs) and “polymer electrolyte membrane base fuel cells” described is this paper, and also referred as PEMFC? The confusion appears at l.35-36 in the introduction.
More generally, the acronyms used throughout the article should be systematically explained.
The paper is well structured. Maybe the § 2.3 should be presented as a new §3, as it depicts the resulting performances of the fuel cells fabricated by 3D printing.
Since the main originality of this review lies in the manufacturing techniques used, in particular 3D printing, it would have been wise to demonstrate more precisely the advantages and disadvantages of each of the techniques described.
§2 is devoted to the description of 3D printing processes. One could expect more than an overview of the different 3D printing processes.
l.117: the § on stereolithography gives interesting details on the characteristics of this process, thicknesses in particular. It is a pity that this precise description is not applied to other processes. A more detailed description could also be based on more precise diagrams than those proposed in Figure 2.
Some limits of the 3D processes are mentioned in the conclusion (l.312 to 315), especially the rheological properties required for each process. These characteristics could have been studied within the description of the advantages and limits of the processes in §2.
l.153: "Tango Plus" appears to be a trade name. Is it possible to have more details on the nature of this photopolymer?
Comments on the Quality of English LanguageThe text needs to be carefully reread to avoid the many typos still present.
In addition, some acronyms have not been defined, making them difficult to read. A list of acronyms would be useful for the reader.
Author Response
Please, find in the attached file point-by-point answers on reviewers' comments

Reviewer 2 Report
Comments and Suggestions for Authors
Comment:
The manuscript of ‘Recent progress 3D printing of polymer electrolyte membrane-based fuel cells for clean energy generation’ is well written. However, major corrections are required to improve the quality of the manuscript.
-Please provide the aim of the review in the last paragraph of the introduction part.
-Grammatical and syntax errors (missing the degree sign – line 284, page 8) are detected in the manuscript. Kindly send for proofreading service.
-Formatting of the manuscript- the citation of the reference shall be written before the full stop. The citations also need to be combined if it is more than 1 e.g., [11-12],[13-14], [15-18].
-The resolution of the figures (1-3) is very poor. Please increase the resolution.
-Please provide the authors opinions on the recent progress 3D printing of polymer electrolyte membrane-based fuel cells
-Please provide the future perspective of 3D printing of polymer electrolyte membrane-based fuel cells

Extensive editing of English language required
Author Response

(The authors gave the same response as above.)

Reviewer 3 Report
Comments and Suggestions for Authors
In this manuscript, the author has succinctly summarized recent advancements in the application of 3D printing for manufacturing various components of hydrogen fuel cells with polymer electrolyte membrane (HFC-PEM), highlighting the salient features of the various 3D printing technologies and emphasising the improvements that have been achieved by the utilisation of 3D printing technology. This topic is quite intriguing and is likely to capture the attention of researchers in the field of energy storage and related areas, including hot topics like energy storage systems and soft robotics that active introduction of machine learning and artificial technologies into the printing process to increase the variability of products. I would like to recommend its publication in Polymers after minor revision of the following concerns:
1. Please unify the capitalization of the first letter in the title.
2. In the introduction, the introduction to 3D printing technology is relatively limited, with only a brief comparison of its advantages and disadvantages compared to traditional preparation methods. It is suggested to expand the background introduction of 3D printing technology in the introduction section, including a brief overview of current status of the application of 3D printing technology in electronic energy storage devices. Additionally, it is recommended for the authors to include more references, such as Chem. Mater. 2023, 35, 5936–5944… And further improve the accuracy and insights of the introduction. This would help demonstrate their comprehensive understanding of the field and enhance the credibility of the article.
3. The text contains numerous grammatical errors, sentence structure issues, and even formatting problems throughout, for example the spelling errors:
“protoтconducting”, “productionnof”, “open’s up”… respectively on line 46, 156, 133, etc. “two electrons and 2 hydrogen ions” on line 45 is structurally inconsistent. The English language in the manuscript requires substantial improvement, ideally by a native speaker, as I struggled to grasp the authors' intended meaning in certain parts of the text.
4. In this manuscript, the author's descriptions of the images are incomplete. For instance, in the case of Fig. (2d) and (2d’), Fig. (3c), (3c’) and (3c’’), the author neither mentions nor elaborates on these figures in the text. Therefore, it is recommended that the author provides additional explanations and descriptions for these figures within the manuscript.
5. In the article, the authors emphasize the simplicity and low cost of the 3D printing method, but there is no corresponding narrative throughout the text to suggest this advantageous convenience over traditional preparation methods. It is recommended that the authors provide a more comprehensive discussion of the simplicity and cost-effectiveness of 3D printing throughout the text to highlight this important point
6. Within this manuscript, the author only introduces three 3D printing methods, and Section 2.1 provides a brief overview of the pros and cons of these three methods, as well as their applicability in fuel cell applications. This narrative might be considered incomplete. Additionally, this section lacks a comprehensive explanation of the working principles, preparation parameters, and materials used in 3D printing technology for battery device fabrication. In comparison to traditional preparation methods, the manuscript does not effectively emphasize the significant role and innovative implications of 3D printing. It is recommended that the author enrich and expand on these aspects in the content for a more comprehensive discussion.
Overall, I would like to recommend its publication after the revision
of above problems.
Comments on the Quality of English Language
Extensive editing of English language required
Author Response

(The authors gave the same response as above.)

Round 2
Reviewer 2 Report
Comments and Suggestions for Authors
The corrections have been incorporated.
Comments on the Quality of English LanguageOk